# Social Media Usage, Working Memory, and Depression: An Experimental Investigation among University Students

**DOI:** 10.3390/bs12010016

**Published:** 2022-01-17

**Authors:** Abeer F. Almarzouki, Renad A. Alghamdi, Roaa Nassar, Reem R. Aljohani, Abdulrahman Nasser, Manar Bawadood, Rawan H. Almalki

**Affiliations:** 1Department of Physiology, Faculty of Medicine, King Abdulaziz University, Jeddah 21589, Saudi Arabia; 2Faculty of Medicine, King Abdulaziz University, Jeddah 21589, Saudi Arabia; renadalghamdi9898@gmail.com (R.A.A.); roaanasser@gmail.com (R.N.); reem.r.johani@gmail.com (R.R.A.); abdulrahman.md.n@gmail.com (A.N.); manarbawadood@gmail.com (M.B.); rawanalmalky52@gmail.com (R.H.A.)

**Keywords:** social media, working memory, depression, anxiety, academic performance

## Abstract

Social media usage (SMU) and its relationship with working memory (WM) and academic performance remain unclear, and there is a lack of experimental evidence. We investigated whether WM mediates the association between SMU and academic performance, including the roles of depression, anxiety, and disordered social media use as possible contributors. A sample of 118 undergraduate students aged 19 to 28 from Saudi Arabia performed a WM test twice; for one assessment, participants were required to interact with social media before the test, and the other test was preceded by painting online. We also measured grade point average (GPA), habitual social media usage (SMU), depression (PHQ-9), anxiety (GAD-7), and disordered social media usage (SMDS). There was no significant difference between WM scores in the social media condition compared to the control condition, but when solely considering at least moderately depressed participants*,* social media use predicted significantly more errors in the social media condition compared to the control condition. Furthermore, higher SMDS scores were significantly predicted by higher PHQ-9 scores and more hours of habitual SMU. GPA scores were not predicted by WM performance or SMU. The present study is one of the first experimental attempts to compare the relationship between SMU and WM and highlights the priming effect of depression on the relationship between SMU and WM.

## 1. Introduction

The use of social media has steadily increased over the past decade. It has become a ubiquitous aspect of everyday living and is rapidly expanding [1]. In a recent study, students have been found to use certain platforms to differing degrees and for different purposes, based on their gender, personality, and age [2]. Furthermore, there was an association between disordered social media usage, specifically on Snapchat, Instagram, and Facebook (among other platforms), and the desire felt by users to socialize and present themselves as more popular [2]. Younger people, in particular, are entering a phase where social media is seen as not only a part of life but also life itself; this perception has dramatically accelerated during and even beyond the COVID-19 pandemic [3]. The literature on this all-pervasive technology is relatively recent, and there is expanding interest among researchers concerning the impact of such technology on society’s general wellbeing. Of particular interest among researchers, parents, and health professionals is the impact of social media on cognitive function [4], educational achievement [5], and mental wellbeing, particularly among the younger generation [6]. However, most work consists of correlation and self-reporting studies and lacks experimental evidence.

The influence of social media usage on cognitive performance is poorly understood. Working memory has garnered particular interest as a cognitive quality that is possibly influenced by social media usage, as it can be one of several predictors of academic achievement [7]. Working memory is the ability to attend to learned information, retain it, and manipulate it to acquire new knowledge, master skills, and direct behavior [8]. The working memory system has a limited capacity, and social media usage may indeed impair working memory by either (1) diverting attention from the learned information, or (2) reducing concentration on the acquired knowledge in favor of the social media content. Both are mechanisms through which working memory performance would be impaired.

Nonetheless, different studies in this field have yielded varying results. Few studies support a positive relationship between more social media usage and higher academic performance [9,10,11], although some studies suggest that media multitasking [12], particularly smartphone usage [13], reduces academic performance. Other studies have not found any differences between those who are and those who are not habitual social media users in relation to working memory [14]. Some studies have found that social media usage leads to improvements in working memory [15]. In a similar pattern, social media has been widely investigated in relation to academic performance, again with conflicting results. Some studies have not found a relationship between academic performance, as measured by GPA, and social media usage [16,17,18]. Others have revealed a negative impact of social media on academic performance; that is, individuals who browse more on social media have worse academic outcomes [19,20,21]. Conflicting findings related to academic performance could be partially attributed to the fact that some studies were based on student opinions rather than objective assessments of grades [22]. In accordance with this statement, Abbas et al. reported that, although students admitted that social media affected their study time negatively, in the opinion of those students, it actually increased their level of scientific knowledge [22]. Another potential reason for such conflicting results might be that previous studies ignored the influence of affect on cognition. It is well known that positive and negative affect can significantly influence prefrontal activation and, thereby, performance in WM tasks [23,24]. Social media usage has also been associated with negative emotional wellbeing [25]. This could contribute to the negative impact of social media on cognitive performance in general or on working memory, specifically [26]. Furthermore, unhealthy (disordered or excessive) use of social media has been found to invariably influence psychological disorders such as depression and anxiety [27].

Methodological limitations could partially explain the variability in the results produced in previous research related to the influence of social media on working memory. First, most studies about academic performance employ cross-sectional and correlational designs or are based solely on self-reports. Second, most research in this area has investigated social media activities without considering the impact of the emotions elicited by social media content and use. Third, most of the literature uses a broad conceptualization of “media multitasking” [12] and smartphone usage [13], and no clear distinction has been drawn between smartphone usage in general and social media per se. Another distinction that has not been clearly defined in a single study is the effect of “acute” use of social media (at a particular moment) and habitual or “chronic” use of social media. In combination, these limitations make it difficult to draw comprehensive conclusions about the influence of social media on working memory or academic performance.

In this study, we designed a novel experimental paradigm to clarify the ambiguity of the effect of social media usage on working memory in previous results by accounting for affective factors (emotional wellbeing, namely, depression and anxiety), patterns of social media use (acute versus habitual use), and behavioral outcomes (academic performance). Our aim in this study is to investigate the following: (1) the impact of acute and habitual usage of social media on working memory performance; (2) the impact of acute and habitual usage of social media on academic performance, as measured by GPA; and (3) the role of negative emotional wellbeing (depression and anxiety) in mediating this association. We hypothesized that participants who used social media before performing a working memory task would demonstrate inferior working memory performance than when they engaged in visual stimuli outside of social media (an active control condition). We also hypothesized that working memory performance would mediate the relationship between social media usage and academic performance, as measured by GPA.

## 2. Materials and Methods

### 2.1. Study Ethics

We designed this study as an experiment-based investigation. The study was performed after obtaining ethical approval from the institutional review board at King Abdulaziz University Hospital (KAUH). The study was conducted from July 2020 until January 2021. The data supporting the findings of this study are available on request from the primary author.

### 2.2. Participants

Participants were recruited by advertising the study, mainly through social media (WhatsApp), in university student communities. All participants were over 18 years old and were university students from major universities in Jeddah, Saudi Arabia: King Abdulaziz University, Jeddah University, Effat University, Ibn Sina National College, Saudi Electronic University, Umm Al-Qura University, Taif University, and Imam Muhammad ibn Saud Islamic University. All participants were free from chronic medical or physiological illness and were living in Jeddah City, the second-largest city in Saudi Arabia. No financial compensation was given to participants. All participants were briefed on the study, gave written informed consent, and were given the right to withdrawal at any time during the study. Participant responses were anonymized.

### 2.3. The Pilot

We first tested the procedures with a small pilot group (*N* = 12; 5 men) with a mean age of 29.1 (range = 32). All participants attended a one-hour session split into three blocks, with ten minutes’ rest between each block. The duration of the first block was 6 min, the second block was 10 min, and the final block was 15 min. For each block, participants would either browse their own social media or participate in other internet-based activities (online painting, news-surfing, or listening to music). All participants completed a computerized working memory task. Based on the findings of this pilot study (for results, see Section 3.2), we proceeded with a larger-scale study that used one block length (6 min) and standardized the nonsocial media task (painting). We also included additional questionnaires, as mentioned below. This time interval was selected as one that seemed to provide a reasonable compromise between being long enough for participants to be engaged in social media yet short enough that participants would not feel bored or tired toward the end of the experiment described below.

### 2.4. Main Study

#### 2.4.1. Participants

We recruited 118 participants: 70 women and 48 men, aged between 19 and 28 years, mean = 23.47, SD = 1.89. Participants were screened over the phone for English fluency and were briefly introduced to the study. Participants were asked to have a good night’s sleep, not take any more than their usual number of stimulants (including nicotine and coffee) and avoid scheduling the in-person study immediately before a stressful event, such as an exam or major deadline. Participants were given instructions to ensure that they understood the tasks required in the study.

#### 2.4.2. Experimental Paradigm

All participants attended a one-hour session, with the testing period split into two blocks; the two blocks were each of 6 min, timed by the researcher, with a rest period between each block. Participants were briefed on the study and given the opportunity to ask questions if needed. The differences between both blocks (social media usage versus painting) were explained to participants and they were asked simply to “perform” the activities in each block, either social media usage or painting, as they would normally do in their free time. After giving their consent, participants were randomly assigned to one of two groups (A/B). Both groups completed the same tasks but in a different order (Figure 1).

Participants first completed a set of demographic questionnaires (age, gender, university, specialty, and year of study). They were then asked to begin the social media block (Group A) or the nonsocial media (painting) block (Group B) for 6 min, timed by the researcher. Immediately following the first block, participants rated how engaging they found the browsing time on a 7-point Likert scale, i.e., to what extent they found the activity they performed to be engaging on a scale from 0 to 7, where 0 meant that they were not engaged at all and 7 meant maximum engagement. Those who responded that they were either completely disengaged (1) or mostly disengaged (2) from either task were excluded from further analysis. Subsequently, participants then performed the spatial working memory (SWM) test (Cambridge Cognition), as described below, to test the effect of the previous block on their working memory. The same sequence was repeated, but participants then completed either the nonsocial media task (painting) block for group A or the social media block for Group B.

After completing the blocks, participants were asked for their smartphone screen time over the last week (total number of hours per week engaged in social media use) as recorded on their smartphones. They were also asked for their academic grade point average (GPA, 5-point scale). Finally, participants were offered rest if desired and were asked to complete four questionnaires: the social media disorder scale (SMDS) [28], media multitasking index (MMI) [29], patient health questionnaire (PHQ)-9 [30] for depression, and general anxiety disorder (GAD)-7 [31] for anxiety. All questionnaires were provided in English and the researcher was available to offer any assistance or clarification of the questions.

#### 2.4.3. The Working Memory Tasks

The CANTAB Spatial Working Memory test measures the ability to remember spatial information and manipulate this information using working memory. The test also assesses the strategy used by participants to remember these items. The test takes about 4 min to complete. Colored squares are displayed on the participant’s screen. In each trial, the task for participants is to figure out which boxes contain blue tokens by initially opening all boxes. This task is supposed to be carried out through a process of elimination and by remembering which boxes are empty. Participants use the blue tokens they find to fill an empty column on the side of the screen. The task is made more complex at each trial “level” by gradually increasing the number of boxes to 12. The position and color of the boxes used are changed at each trial. Selecting those boxes that have already been found to be empty and rechoosing boxes that participants have already found to contain a token are considered errors. The error number and the strategy used by participants to identify the token locations constitute the outcome measures.

### 2.5. Data Analysis

All data were entered into SPSS Version 26. The researcher screened for the presence of missing values and data-entry errors. Normality tests were conducted on all continuous variables to determine which descriptive or inferential statistics were appropriate for analyzing those variables. For normally distributed variables, the measures of central tendency to means and measures of dispersion were standard deviations, ranges, and interquartile ranges. For non-normally distributed variables, the measure of central tendency was the median, and the measure of dispersion was the interquartile range.

Dependent variables in regression modeling included the total number of errors and the strategy score, with a lower strategy score indicating a more efficient strategy. Trials with additional tokens were more difficult, which is why the number of errors made in each of the trials with 6, 8, or 12 tokens was compared with baseline difficulty (4 tokens) to select sufficiently difficult trials. For each, the error differential (e.g., total errors in 6-token trials vs. 4-token trials) was calculated for each participant. A 95% confidence interval (CI) was estimated using Monte Carlo simulations over 1000 randomized permutations of each distribution. The easiest difficulty (fewest tokens), which did not include zero in its CI relative to the baseline, represented a trial that was difficult enough to pose a challenge to participants without hindering performance by being unduly difficult. Total errors in trials with this number of tokens and strategy scores for trials up to and including this number of tokens were taken as dependent variables.

For the pilot study, performance was evaluated as the total number of errors that a participant made during the 6-, 10-, or 15-minute variants of each trial type (social media or nonsocial media). Paired sample *t*-tests were used to compare the number of errors made between time intervals and between trial types. The best trial length was taken as the shortest length of time that elicited a difference in the number of errors between trial types.

Independent variables included either overall social media use (hours per week) or SMDS score [28], test conditions (acute social media use or painting), an interaction between the two, and independent variables including participant age, gender, PHQ-9 [30], and GAD-7 [31] scores. Normality of residuals was ensured using QQ-plots.

For each type of media consumption, responses greater than three standard deviations beyond the mean were identified as outliers and were excluded for each of print media (n = 2), TV (n = 2), computer-based video (n = 6), music (n = 6), nonmusical audio (n = 2), mobile phone video calls (n = 6), email (n = 2), web surfing (n = 4), instant messaging (n = 6), and other computer applications (n = 6). Comparison of participant characteristics between groups was carried out using *t*-tests with independent samples. Correlations among numeric measures and scales of interest were calculated using the Pearson correlation coefficient. Regression modeling was completed using linear mixed modeling (package lme4 in R, version 3.6.2), accounting for repeated measures among participants by including a random grouping factor of each participant.

## 3. Results

### 3.1. Participants

Complete data were collected from the 118 participants. Characteristics for these participants are provided in Table 1.

In total, 1 participant was excluded for not understanding the instructions after completing the task, and 11 were excluded due to the reporting being either mostly or completely disengaged from both tasks. The final number of participants was therefore 106. Group characteristics are outlined in Table 2. Despite random group assignment, significant differences between groups were found in GAD-7 and PHQ-9 scores. These items were controlled for in all models, as described above, to delineate true experimental effects from this potential contradiction. Participant age was also included in all models. Social media use did not significantly differ between groups, either overall or in any specific category.

### 3.2. Pilot Study

All three blocks showed a significant increase in error rate from the social media task compared to the nonsocial media task (t (6) = 4.87, *p* = 0.003), suggesting a possible reduction of working memory functioning following the social media usage test.

### 3.3. Spatial Working Memory (SWM) Performance

Neither 6-token (95% CI = −1.63 to 9.23 more errors) nor 8-token trials (95% CI = −1.80 to 17.25 more errors) were significantly more challenging than baseline difficulty (4-token trials). Therefore, we used total errors during 12-token trials (95% CI = 0.03 to 53.21) and strategy scores were selected for 6, 8, or 12-token trials, due to the program’s way of reporting this variable. Accordingly, our dependent variables were “error” and “strategy” variables.

The number of errors (mean and standard deviation) made in the spatial working memory (SWM) task during each condition are outlined in Table 3. Notably, the number of errors increased as the number of tokens in a trial increased. Regardless of the number of tokens, there was no significant difference between the number of errors made in the “social media” condition relative to the “painting” condition. As noted above, the order of conditions was controlled for. We also compared SWM strategy scores for trials with six or eight tokens against the SWM Adult Norms bank v1.1.

### 3.4. Measure Correlations

Pearson correlations were investigated across the scales and measures of interest, as outlined in Table 4. Briefly, the number of errors in the SWM task was moderately correlated with the SWM strategy (a lower strategy score is better). SMDS scores were weakly correlated with each score of GAD-7 (r = 0.359) and PHQ-9 (r = 0.444), while GAD-7 and PHQ-9 scores were strongly correlated (r = 0.818). Neither GPA nor overall social media use showed considerable correlations with any other measure.

### 3.5. Statistical Modeling

When evaluating the impact of social media use through the weekly number of hours spent using social media, the total number of errors in the SWM task was not affected by hours of social media use (F (1,99) = 1.193, *p* = 0.277), experimental conditions (painting or social media trial conditions) (F (1,103) = 0.092, *p* = 0.762), or an interaction between the two (F (1,103) = 0.139, *p* = 0.709). When considering disordered social media use, there was similarly no significant effect on the SMDS score (F (1,99) = 0.194, *p* = 0.660), experimental conditions (F (1,103) = 0.067, *p* = 0.796), or an interaction between the two (F (1,103) = 0.125, *p* = 0.725) on the number of errors recorded in the SWM task. There was no impact on either model of age, sex, GAD-7, or PHQ-9 scores.

In addition to the number of errors, we investigated strategy scores (a lower score is better). In terms of the number of hours per week engaged in social media use, strategy was not affected by overall social media use (F (1,100) = 0.106, *p* = 0.745), experimental conditions (F (1,103) = 0.287, *p* = 0.593), or their interaction (F (1,103) = 0.810, *p* = 0.370). Strategy was also not affected by SMDS scores (F (1,100) = 1.011, *p* = 0.452), experimental conditions (F (1,103) = 0.452, *p* = 0.452), or their interaction (F (1,103) = 0.365, *p* = 0.547). There was a significant effect of gender on strategy (F (1,100) = 10.679, *p* = 0.001) with men showing on average a strategy score of 9.88 (SE = 0.76) and women a score of 13.22 (SE = 0.62). There was no effect on age, GAD-7, or PHQ-9 scores.

We were interested to observe whether the above effects might be limited to only those participants who were at least moderately depressed (PHQ-9 scores of at least 10, n = 48, 23% of our sample) or who had at least moderate anxiety (GAD-7 scores of at least 10, n = 38, 18% of our sample). For this sub-analysis only, we dichotomized the sample, based on the diagnostic cut-offs for the chosen questionnaires. We tested whether the total number of errors or SWM strategy scores were associated with the experimental conditions, social media use (either defined as the total number of hours of social media use or SMDS scores, one model for each), or an interaction between the two, while controlling for age, sex, GAD-7, and PHQ-9 scores. When considering only those participants who were at least moderately depressed, the social media use condition predicted significantly more errors than the control condition (F (1,22) = 4.52, *p* = 0.045)**,** whereas SMDS scores were not related to the number of errors, and there was no interaction between the two. In addition, while the social media use condition was predictive of the number of errors for these participants, it was not predictive of SWM strategy. Even while controlling for strategy, the social media use condition remained significantly associated with errors in this subset (F (1,22) = 6.72, *p* = 0.017). Looking at the subset of those with at least moderate anxiety, neither the number of errors nor strategy was significantly associated with the experimental conditions, social media use (in terms of hours or SMDS scores), or an interaction between both.

### 3.6. Social Media Activities

We investigated whether the time spent on any specific activities related to social media was related to either SWM performance, in terms of error or strategy, or GPA. All these analyses were controlled for participant age, gender, GAD-7, and PHQ-9 scores. There was no significant effect on conditions and no interaction between conditions and web surfing in predicting the number of errors. In addition, none of the web-surfing behaviors, conditions, or interactions predicted SWM strategy. No other specific activities were related to either the number of errors or strategy in the SWM task. In addition, no specific activities were significantly related to GPA. Lastly, when considering only those who were at least moderately depressed, or only those who had at least moderate anxiety, no measures of social media use were related to GPA.

### 3.7. Social Media Use and Mental Wellbeing

Whether social media use affects mental wellbeing, as assessed using the GAD-7 and PHQ-9, was also considered. While there was no significant association identified between weekly hours of social media use and either of these measures, GAD-7 scores were found to be significantly higher in those with higher SMDS scores (t (102) = 3.252, *p* = 0.002), as were PHQ-9 scores (t (102) = 4.355, *p* < 0.001), while controlling for age and gender. These findings are shown in Figure 2.

Given the relationship between GAD-7, PHQ-9, and SMDS scores, we investigated their association in more depth. A linear model was used to test whether SMDS scores were predicted by GAD-7 and PHQ-9 scores while controlling for total social media use, age, and gender. In this model, higher SMDS scores were significantly predicted by higher PHQ-9 scores (t (100) = 2.5781, *p* = 0.012) and by more total hours engaged in browsing social media (t (100) = 2.998, *p* = 0.003). However, GAD-7 scores showed no significant association in this model, suggesting that their relationship with SMDS scores is better accounted for by their correlation with PHQ-9 scores.

## 4. Discussion

The impact of social media usage among young people remains under-investigated, with many studies relying on correlational studies and cross-sectional designs. Furthermore, most studies have investigated social media usage in isolation from mental wellbeing, which has been proven to be a factor influencing working memory and social media usage. Therefore, the current study aimed to expand prior research linking social media usage and working memory failures by implementing a novel paradigm to investigate this relationship more robustly. It was hypothesized that participants who used social media before performing a working memory task (the trial condition) would demonstrate poorer working memory performance than when they had not used social media (the control condition). However, there was no significant difference in working memory performance between the two conditions. Furthermore, academic performance, as measured by GPA and working memory, was not predicted by social media use (whether acute or habitual). Nevertheless, in a subset of at least moderately depressed individuals, acute social media usage significantly amplified errors of working memory.

### 4.1. Working Memory and Social Media Usage

Our results indicate that working memory may be resilient to social media usage, at least for a healthy group of adults. There are several reasons why SM and non-SM activities did not differently affect the participants’ performance of the WM tasks in our study. First, although it could be argued that both the painting and the SM-surfing activities were equally engaging, when we assessed the participants’ emotional engagement after each block, it was clear from subjective ratings that there was greater engagement in the social media task than the painting task. Second, we attempted to incorporate enough exposure to social media to be able to see its impact on working memory performance by experimenting with different durations in the pilot study. This revealed an effect on social media usage even among a small number of participants during the time frame selected. We do acknowledge that it remains challenging to create experimental conditions that can mimic the everyday use of social media. However, in accordance with other studies, we also measured the participants’ habitual use of SM, based on their screen-time records of the previous week, rather than solely on acute exposure. However, we could not find an association between the habitual usage of social media, as recorded by screen time, and working memory or academic performance. In earlier research, Doss et al. used a daily diary to track social media usage and working memory and found that working memory and negative affect varied on a daily basis. On those days when participants reported a higher negative impact, they scored worse in assessments of working memory [32]. A daily follow-up of SMU and WM performance using a journal, following the procedure of Doss et al. [32], may be a more accurate record of this association.

Interestingly, the results showed a difference between the conditions, regarding those who were at least moderately depressed, since they made more errors in the WM task after acute exposure to social media. Affective cognitive processing is a determinant of human behavior. Actions and judgments occur in an emotional framework, and accordingly, cognitive functions such as working memory are shaped by an individual’s emotional state [33]. Emotional engagement or “sensitivity” to environmental stimuli updates working memory conentuosely and promotes the process of learning and development [34]. Research concerning mood disorders suggests that “emotional” cognition, specifically that related to working memory, is impaired by depression [34]. Even in non-emotional tasks like the SWM, individuals with depression seem to make more between-search errors (selecting boxes that have already been found to contain a token) with an increased difficulty level of tasks and use WM strategies less effectively [35]. Furthermore, participants with sub-clinical but higher depression scores than the control group performed significantly more poorly on the strategy dimension [36].

A recent study by Zhang et al. [37] investigated deficits in WM updating, specifically those related to positive information, in major depressive disorder patients. The authors uncovered a positive-specific impairment in this group of patients, suggesting that they are insensitive to positive cues in the early encoding phase of WM updating [37]. In our study, identifying the correct tokens and moving to the next level in the WM task might imply a reward value. It could be that those who are at least moderately depressed have a blunted response to this positive information. To measure this finding robustly, however, we would need to explore the relationship in a sample of individuals with clinically diagnosed depression.

Another possibility explaining why this particular group performed worse on WM after social media exposure could be that acute SMU in this experiment was an “affective” drain on the limited-capacity working memory and that depressed individuals are more vulnerable to such a drain than healthy individuals, resulting in performances lower than those of the control group in working memory tasks. Previous research on working memory training among anxiety patients has resulted in improved working memory capacity post-training, and this WM gain was also significantly correlated with anxiety reduction [38]. Although, in our study, we did not find poorer working memory performance associated with anxiety but rather with depression, the findings of this study do suggest that emotionally vulnerable individuals could indeed have limited working memory capacity, which is fortunately trainable.

### 4.2. Academic Performance and Social Media Usage

Working memory performance and social media usage (acute and habitual) did not predict GPA in our study. Several previous studies reported similar findings [16,17,18], while others found a positive or negative relationship, as previously discussed. Nonetheless, there is enormous variability between the measures that were used for academic performance assessment, including the number of hours spent on social media [39] as well as social media assessment (self-reported questions) [39]. Our study assessed academic performance using GPA and smartphone screen records of hours spent on social media as objective measures, rather than relying on student reports. In light of such work, our study and similar works have introduced the notion that GPA may be contingent on changes driven by social media or working memory; it is highly possible that the higher cognitive value of maintaining a good GPA outweighs the cognitive load that could be potentially drained by social media usage, and that students are purposefully and selectively keeping their resources directed to the higher-value target (a good GPA). Research in the area suggests that GPA scores may not be a sensitive indicator of social media changes in working memory. However, this finding should be approached with caution, and longitudinal studies are needed to explore the question further.

### 4.3. SMDS Scores

Our study also found that higher SMDS scores were significantly predicted by higher PHQ-9 scores and more hours being spent engaging with social media. However, the relationship between SMDS and depression was much stronger than that between SMDS and general SMU. It has been previously found that excessive social media usage is strongly correlated with the development of addictive behaviors that negatively impact emotional wellbeing. This correlation has been previously investigated by van den Eijnden et al. [6] in an attempt to include social media disordered usage in the *Diagnostic and Statistical Manual of Mental Disorders* [40] classification system of addictive disorders. The pattern found in this study could be used to explore whether lower scores on the SMDS questionnaire reflect a resilience to the impact of SMU on working memory [28]. Although the underlying mechanism of how social media usage drives addictive behavior or vice versa is not understood, the two are clearly linked. This finding requires urgent attention from health and educational professionals, given that younger generations are engaging in social media more frequently at a time of life when their personality and psychological wellbeing are still developing.

### 4.4. Limitations

First, the sample in this study might be biased because it was selected from people who responded to a social media advertisement, and the cohort consisted of university students, presenting a narrow population. We did not investigate patterns in social media usage, such as “likes”, posts on walls, sharing or retweeting, status updates, photo uploads, videos, or comments on posts. In their earlier research, Alloway et al. [41] also found that passive platforms such as YouTube did not significantly impact working memory; hence, it could be a fact that interactive engagement with platforms is essential for the effect to be reflected in working memory performance. While the social media task did receive a higher engagement rating than the painting task, this speaks more toward the need to find an equally engaging task for the control conditions in subsequent studies. Additionally, the duration and timing of the social media task used in this study might not adequately represent students’ everyday use of SM or be sufficient to influence working memory performance. Furthermore, as noted in the comparison with the study by Doss et al. [32], it is possible that habitual use, as measured in one screenshot setting (as we have done in this study), may not be a true reflection of everyday social media usage and, hence, may not capture its true effect on working memory. Lastly, differences in academic achievement might not have been detected because the sample was quite homogeneous in terms of GPA, as reflected by the high mean GPAs and the low SDs.

### 4.5. Future Work

Our study is among the first seeking experimental evidence to investigate the complex relationship between social media usage and cognitive function, specifically working memory and academic performance. Though there are some limitations to the experimental designs that have been discussed, the work we conducted suggests that depressed or problematic users should be cautious about how they use such technologies. Future studies are necessary to fully understand the causality and underlying mechanisms of the relationship between social media, working memory, and academic achievement, possibly by implementing an experimental longitudinal design.

## Figures and Tables

**Figure 1 behavsci-12-00016-f001:**
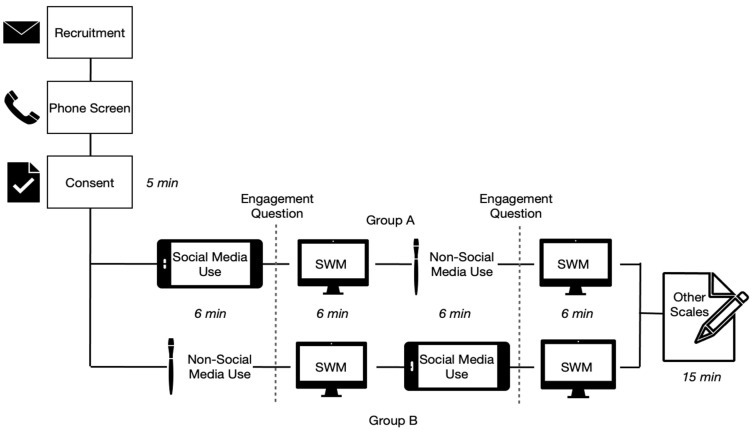
Experimental setup of the study, including the recruitment process and randomization of groups.

**Figure 2 behavsci-12-00016-f002:**
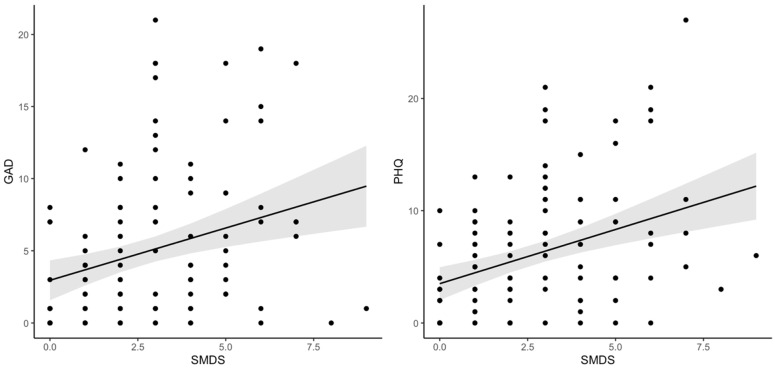
PHQ-9 and GAD-7 scores in relation to SMDS scores.

**Table 1 behavsci-12-00016-t001:** Breakdown of participant demographics.

	Number	Percent
**Total Sample**	118	—
**Gender**		
Women	70	59%
Men	48	40%
**Faculty**		
Economics and Administration	15	13%
Health Sciences	14	12%
Humanities and Social science	7	6%
IT and Engineering	18	15%
Medicine	59	50%
Science	4	3%
Other	1	1%
**Program year**		
Intern	2	2%
First	6	5%
Second	10	8%
Third	19	16%
Fourth	18	15%
Fifth	11	9%
Sixth	52	44%

**Table 2 behavsci-12-00016-t002:** Participant characteristics by group, with overall and between-group significance testing. Asterisks denote significant differences between groups. Social media types are assumed to be non-independent, and FDR-adjusted *p*-values are reported for these items. Statistical significance is denoted using asterisks (*, *p* < 0.05).

	Group		Overall	Significance
	A (n = 54)	B (n = 52)	(N = 106)	
Sex (f, %)	36 (66.67%)	27 (51.92%)	63 (59.43%)	*X*2 = 1.82, *p* = 0.178
Age	23.4 (1.66)	22.78 (1.9)	23.09 (1.8)	t (101) = 1.79, *p* = 0.076, d = 0.348
Social media	24.33 (14.18)	28.37 (17.25)	26.32 (15.82)	t (99) = −1.31, *p* = 0.192, d = 0.256
Print media	1.3 (2.6)	0.95 (1.74)	1.13 (2.22)	t (93) = 0.81, *p* = 0.697, d = 0.158
TV	3.87 (8.15)	3.66 (6.4)	3.77 (7.32)	t (100) = 0.15, *p* = 0.938, d = 0.029
Computer-based video	5.47 (5.16)	5.57 (7.45)	5.52 (6.34)	t (87) = −0.08, *p* = 0.938, d = 0.016
Music	6.68 (7.52)	5.08 (7.27)	5.9 (7.41)	t (103) = 1.11, *p* = 0.542, d = 0.216
Nonmusical audio	1.62 (2.55)	0.93 (1.5)	1.29 (2.12)	t (87) = 1.69, *p* = 0.235, d = 0.330
Mobile phone video call	2.78 (3.96)	2.28 (3.95)	2.53 (3.94)	t (101) = 0.65, *p* = 0.743, d = 0.126
Email	0.78 (0.84)	0.5 (0.67)	0.64 (0.77)	t (99) = 1.94, *p* = 0.235, d = 0.369
Web surfing	7.82 (8.38)	4.47 (5.99)	6.18 (7.46)	t (94) = 2.35, *p* = 0.210, d = 0.459
Instant messaging	7.97 (8.54)	7.07 (9.73)	7.53 (9.11)	t (99) = 0.5, *p* = 0.771, d = 0.098
Other computer applications	3.27 (4.63)	1.78 (4.09)	2.53 (4.41)	t (100) = 1.72, *p* = 0.235, d = 0.341
SMDS	2.69 (1.8)	2.77 (2.45)	2.73 (2.13)	t (94) = −0.20, *p* = 0.841, d = 0.037
GAD-7	6.28 (5.44)	3.79 (4.95)	5.06 (5.33)	t (104) = 2.46, *p* = 0.015, d = 0.497 *
PHQ-9	7.85 (5.71)	4.63 (5.34)	6.27 (5.74)	t (104) = 2.99, *p* = 0.003, d = 0.583 *
GPA (5)	4.20 (0.55)	4.30 (0.40)	4.25 (0.50)	t (95) = −1.50, *p* = 0.136, d = 0.208

**Table 3 behavsci-12-00016-t003:** Average and standard deviation for the number of errors in each condition for trials containing different numbers of tokens, with independent samples t-tests testing the significance of the performance difference.

	Painting	Social Media	Difference
4 Tokens	0.43 (1.16)	0.49 (1.1)	t (209) = −0.36, *p* = 0.716
6 Tokens	1.97 (3.05)	2.17 (3.43)	t (207) = −0.44, *p* = 0.657
8 Tokens	5.32 (6.45)	5.42 (6.64)	t (210) = −0.1, *p* = 0.917
12 Tokens	23.87 (16.84)	24.07 (16.25)	t (209) = −0.09, *p* = 0.93

**Table 4 behavsci-12-00016-t004:** Pearson correlation coefficients among scales and measures of interest. Statistical significance is denoted using asterisks (*, *p* < 0.05).

	GPA	Social Media	SMDS	GAD	PHQ-9	SWM Error	SWM Strategy
GPA	1.000	0.039	0.064	0.122	−0.070	0.002	0.034
Social media	0.039	1.000	0.243	0.088	−0.025	−0.090	0.008
SMDS	0.064	**0.243 ***	1.000	0.359	**0.444 ***	0.082	0.213
GAD	−0.122	−0.088	**0.359 ***	1.000	**0.818 ***	0.109	0.236
PHQ	−0.070	−0.025	**0.444 ***	**0.81 ***	1.000	0.079	0.231
SWM Error	0.002	−0.090	0.082	0.109	0.079	1.000	0.605
SWM Strategy	0.034	0.008	0.213	0.236	**0.231 ***	**0.605 ***	1.000

## Data Availability

Data available upon request due to ethical restrictions.

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
