# Peer review of "Social Media Usage, Working Memory, and Depression: An Experimental Investigation among University Students"

_behavsci, 2022, doi:10.3390/bs12010016_

Round 1

Reviewer 1 Report

Overview:
Very well-written study with high importance in today’s behavioral and cognitive sciences. 

Abstract:  Good overview of the context of the study, methods, population, hypothesis, and general conclusions.  More specific details should be included as to the demographics of the population (what country/cities in addition to university and age)

Introduction:  Overall very good review of existing literature with a critical approach to looking at what exists, bias, and what is lacking.

Line 35 – the word invasive is a bit subjective.  Consider using another adjective or clarify the term.

Materials and Methods:  Overall a thorough description of what was done and the intended analysis.  Some additional information is needed regarding the participant population such as gender and specific university/universities. 

I’m glad to see the authors included their pilot study into the paper. 

The participant section needs to include gender of participants. 

Results: 

The information from genders needs to also be included in the participant section of the Methods.

Discussion:  Good overview of findings, limitations, and how they fit with the existing literature. 

Author Response

Reviewer 1

Overview: Very well-written study with high importance in today’s behavioral and cognitive sciences. 

Response: We would like to thank the reviewer for taking time to review our manuscript and for highlighting these strengths of our study.

Abstract: Good overview of the context of the study, methods, population, hypothesis, and general conclusions. More specific details should be included as to the demographics of the population (what country/cities in addition to university and age)

Response: We would like to thank the reviewer for these comments. We have added more details in the abstract regarding the demographics of the population (age, country) but due to word limit we added university names in the method section. We hope this is acceptable.

Introduction:  Overall very good review of existing literature with a critical approach to looking at what exists, bias, and what is lacking.

Line 35 – the word invasive is a bit subjective.  Consider using another adjective or clarify the term.

Response: We would like to thank the reviewer for this comment. The word “invasive” has been changed to “all-pervasive”.  Line 35 - The literature on this all-pervasive technology is relatively recent, and there is expanding interest among researchers concerning the impact of such technology on society’s general wellbeing.

Materials and Methods:  Overall a thorough description of what was done and the intended analysis.  Some additional information is needed regarding the participant population such as gender and specific university/universities. I’m glad to see the authors included their pilot study into the paper. The participant section needs to include gender of participants. 

Results: The information from genders needs to also be included in the participant section of the Methods.

Response: We would like to thank the reviewer for these comments. We have added more details in the participants section regarding the demographics of the population (age, gender, country and university).

Discussion:  Good overview of findings, limitations, and how they fit with the existing literature. 

Response: We would like to thank the reviewer for highlighting these strengths of the discussion.

Reviewer 2 Report

Dear Authors,

The topic is of interest and the research is relevant for the journal.

There are however a few suggestions I would like to make:

  1. include the period in which the research has been conducted;
  2. describe the exact activities assigned to the two groups;
  3. check language use once more since I have had difficulties to read and comprehend the manuscript, particularly section 2 (announcing the topics that are to be developed as well as going from general to particular might be of help);
  4. check the layout once more, particularly tables that start on one page and continue on the next one.

Good luck!

The Reviewer

Author Response

Reviewer 2

Response: We would like to thank the reviewer for taking time to review our manuscript and for highlighting these strengths of our study.

  1. describe the exact activities assigned to the two groups;

Response: We would like to thank the reviewer for this comment. We agree with the reviewer’s comment on this point, and we have added more details in the method section exact activities assigned to the two groups.

  1. check language use once more since I have had difficulties to read and comprehend the manuscript, particularly section 2 (announcing the topics that are to be developed as well as going from general to particular might be of help);

Response: We would like to thank the reviewer for this comment. Although the article had already been proofread prior to submission, we may have missed some errors. Per the reviewer’s comments, we have now had the article professionally proofread for a second time by MDPI Service.

  1. check the layout once more, particularly tables that start on one page and continue on the next one.

Response: We would like to thank the reviewer for taking time to review our manuscript. Although the article had already been proofread prior to submission, we may have missed some errors. Per the reviewer’s comments, we have now had the article professionally checked for a second time by MDPI Service.